# A Data-Driven Approach to Link GC-MS and LC-MS with Sensory Attributes of Chicken Bouillon with Added Yeast-Derived Flavor Products in a Combined Prediction Model

**DOI:** 10.3390/metabo15050317

**Published:** 2025-05-08

**Authors:** Simon Leygeber, Carmen Diez-Simon, Justus L. Großmann, Anne-Charlotte Dubbelman, Amy C. Harms, Johan A. Westerhuis, Doris M. Jacobs, Peter W. Lindenburg, Margriet M. W. B. Hendriks, Brenda C. H. Ammerlaan, Marco A. van den Berg, Rudi van Doorn, Roland Mumm, Age K. Smilde, Robert D. Hall, Thomas Hankemeier

**Affiliations:** 1Leiden Academic Centre for Drug Research, Leiden University, Einsteinweg 55, 2333 CC Leiden, The Netherlands; 2Laboratory of Plant Physiology, Wageningen University and Research, Droevendaalsesteeg 1, 6708 PB Wageningen, The Netherlands; 3Swammerdam Institute for Life Sciences, University of Amsterdam, Science Park 904, 1098 XH Amsterdam, The Netherlands; 4Unilever Foods Innovation Centre, Bronland 14, 6708 WH Wageningen, The Netherlands; 5Leiden Center for Applied Bioscience, University of Applied Sciences Leiden, Zernikedreef 11, 2333 CK Leiden, The Netherlands; 6Science & Research, DSM-Firmenich, Alexander Fleminglaan 1, 2613 AX Delft, The Netherlands; 7Taste, Texture & Health, DSM-Firmenich, Alexander Fleminglaan 1, 2613 AX Delft, The Netherlands; 8Wageningen Plant Research (Bioscience), Wageningen University and Research, Droevendaalsesteeg 1, 6708 PB Wageningen, The Netherlands

**Keywords:** stir bar sorptive extraction–gas chromatography–mass spectrometry (SBSE-GC-MS), HILIC, RPLC, sensory analysis, savory foods, yeast-derived flavor products, chicken bouillon, flavor prediction modeling

## Abstract

Background: There is a continuous demand to create new, superior sensory food experiences. In the food industry, yeast-derived flavor products (YPs) are often used as ingredients in foods to create new aromas and taste qualities that are appreciated by consumers. Methods: Chicken bouillon samples containing diverse YPs were chemically and sensorially characterized using statistical multivariate analyses. The sensory evaluation was performed using quantitative descriptive analysis (QDA) by trained panelists. Thirty-four sensory attributes were scored, including odor, flavor, mouthfeel, aftertaste and afterfeel. Untargeted metabolomic profiles were obtained using stir bar sorptive extraction (SBSE) coupled to GC-MS, RPLC-MS and targeted HILIC-MS. Results: In total, 261 volatiles were detected using GC-MS, from chemical groups of predominantly aldehydes, esters, pyrazines and ketones. Random Forest (RF) modeling revealed volatiles associated with roast odor (2-ethyl-5-methyl pyrazine, 2,3,5-trimethyl-6-isopentyl pyrazine) and chicken odor (2,4-nonadienal, 2,4-decadienal, 2-acetyl furan), which could be predicted by our combined model with R^2^ > 0.5. In total, 2305 non-volatiles were detected for RPLC-MS and 34 for targeted HILIC-MS, where fructose-isoleucine and cyclo-leucine-proline were found to correlate with roast flavor and odor. Furthermore, a list of metabolites (glutamate, monophosphates, methionyl-leucine) was linked to umami-related flavor. This study describes a straightforward data-driven approach for studying foods with added YPs to identify flavor-impacting correlations between molecular composition and sensory perception. It also highlights limitations and preconditions for good prediction models. Overall, this study emphasizes a matrix-based approach for the prediction of food taste, which can be used to analyze foods for targeted flavor design or quality control.

## 1. Introduction

In food studies, it is of particular interest to understand the relationships between the perceived sensory attributes and the chemical composition of food products better. Associating (non-)volatile compounds with odor or flavor qualities is important with respect to food product design and formulation. If certain volatile compounds have a strong correlation with a desired odor/flavor attribute, the composition and processing procedures of a food product can be improved through the use of this knowledge. For the industry, formulating new products with superior sensory qualities is of great relevance. Savory ingredients are widely used as constituents of many food products. For example, yeast-derived ingredients are often added to ready-to-eat soups and meals, bouillons and snacks. For many years, researchers and food designers have been trying to relate sensory data and chemical data in food to evaluate, monitor and improve food processing strategies and food compositions. Food processing is a complex procedure that involves many chemical reactions that lead to the formation of a vast array of molecules capable of impacting the overall flavor of a product [1]. The most common reactions occurring during food processing are lipid oxidation and Maillard reactions [2]. The latter occur between amino acids and reducing sugars, triggering the formation of many small flavor molecules including both volatile and non-volatile compounds. Monitoring food processing reactions should help the industry and scientists gain a more mechanistic understanding of the steps involved and allow us to select processes that contribute best to the formation of desirable flavor attributes, while importantly also avoiding the formation of off-flavors. There have been regular attempts to associate certain odor properties with specific volatile compounds [3]. For instance, wine is one of the most common food products studied to link desired aromas with the presence of certain volatile aroma compounds. Negative smoke taint attributes (‘ashiness’, ‘greenness’ and ‘burnt rubber’), for example, were found to be associated with the presence of volatile phenols [4]. However, the same study also found an increased taint perception with high levels of guaiacol and low levels of phenols. This suggested that multiple molecules can be associated with a sensory attribute, showing that correlations are usually difficult to interpret and are frequently attributed to the generally complex nature of flavor [3].

In the study described here, we focused on the most-used savory ingredients in the food industry: yeast (*Saccharomyces cerevisiae*)-derived flavor products (YPs). The use of YPs offers several advantages [5], including the potential to replace artificial flavorings while enhancing flavor complexity and balance. During the production of YPs, combinations of variant fermentation protocols, downstream processing and Maillard reactions—typical for food processing—result in an extended diversity in chemical compounds and flavors that could be used to make food more palatable. YPs can positively enhance meaty aromas, thereby making the use of meat extracts or artificial additives redundant. This has been the topic of several studies, and results have shown, for example, that especially for chicken and pork flavors, chemical reactions between cysteine and reducing sugars (both can be found in yeast extracts) were correlated with sensory impact [6]. Volatile compounds including 2-methyl-3-furanthiol, 2-furfurylthiol, methionol, 2,4,5-trimethyl-thiazole, nonanol and 2-nonenal have been proposed as being important for the specific flavor of chicken [6,7]. In addition, alkylpyrazines and trithiolanes have been associated with fried chicken and roasted chicken, but not with chicken broth [6]. This suggests that many volatile groups (or a combination of them) can contribute to overall sensory perception.

The modeling of the intrinsic non-linear relations between data on sensory perception and data from instrumental measurements of chemical composition has been the scope of many food studies. Regression analyses, such as partial least squares analysis (PLS), are the most used statistical strategies to find associations between sensory and instrumental data [8]. However, other tools like Random Forest and Elastic net have also been used [9]. In an earlier related pilot study, where tomato soups with added YPs were tested, PLS-based strategies for predicting certain taste attributes for odor and flavor or specific ingredients were evaluated. This pilot study revealed that prediction models struggle when samples exhibit limited sensory variation, highlighting the need for broader compositional diversity [10].

To continue the study of the sensory impact of YPs, a broader range of YPs, having more diversity in sensory profiles and chemical composition, was applied in a simpler matrix (chicken bouillon instead of tomato soup) to better expose the YPs’ contribution. SBSE-GC-MS untargeted analysis was used for this diverse dataset since it was able to extract a wide range of volatile groups, including the less volatile and less apolar compounds. In a similar matter, untargeted RPLC-MS was used for non-volatile compounds, and targeted HILIC-MS was used to extend the metabolomic coverage for sugars/polar compounds.

The goal of the present study was to identify relationships between metabolites and sensory attributes with the aim of developing predictive models that should drive future food re-formulation. To that end, we combined the outcome of a comprehensive sensory evaluation by quantitative descriptive analysis (QDA) and the untargeted volatile profiles obtained by SBSE-GC-MS, untargeted RPLC-MS and targeted HILIC-MS of chicken bouillon samples prepared with the addition of diverse YPs. Using a machine learning algorithm (RF) we aimed to correlate specific volatile compounds with individual odor attributes and non-volatile compounds with flavor attributes and set up the fundamentals for obtaining good prediction models that can be used for future chicken-flavored food products. This study represents an extensive effort combining a professional sensory panel with multi-platform metabolomics to investigate the complex flavor profiles of bouillons with compositional variations.

## 2. Materials and Methods

### 2.1. Food Materials and Bouillon Preparation

The samples used for the sensory and chemical analysis were prepared by using commercial chicken bouillon powders that were then supplemented with YPs (provided by DSM-Firmenich, Delft, The Netherlands). The final composition of the bouillon samples was varied by adding different yeast products and by combining these with different dosages of the base chicken bouillon, compared to commercial formulations (high = 80%, middle = 40%, low = 0%). Seventy-one bouillon samples were prepared and analyzed, each of which contained one of thirty-six different YPs, each with two different chicken aroma dosages (for the experimental design, see Appendix A). YPs were categorized into four types: process flavor, process flavor blend, yeast extract and yeast extract blend. The bouillons were prepared by stirring the bouillon powders into 1 L hot water (90 °C). The bouillons were gently simmered for 5 min and occasionally stirred. After that, the majority was administered to a trained sensory panel for sensory evaluation, while small aliquots (20 mL) were transferred into glass vials that were stored in a freezer (at −80 °C) until further chemical analysis.

### 2.2. Sensory Evaluation

Sensory evaluations were performed at the laboratories of DSM-Firmenich using established standard protocols following ISO 6658:2017 [11]. An extended sensory evaluation of the samples was performed by applying QDA. QDA investigates the odor (aroma), flavor, mouthfeel, aftertaste and afterfeel attributes, such as odor intensity, yeast odor, umami flavor, chicken flavor, fatty mouthfeel, salt aftertaste and mouthcoating afterfeel of a set of prepared chicken bouillons (See Table 1). In total, thirty-four attributes were assessed by a group of experienced, trained (>10 years) panelists (n = 15). All panel members were tested and selected based on sensory sensitivity, discriminative and descriptive ability [12]. Prior to the actual experimental sessions, two training sessions were organized to familiarize the panel members with the specifics of the study products and to align on the relevant attributes. The sensory evaluation was divided into eight sessions. For each session, ten samples were tested by each panelist. All bouillon samples were kept in a holding cabinet at 60 °C, and 50 mL aliquots of warm bouillon were served one by one to the panelists in a white polystyrene cup. During the QDA measurement, the intensities of the attributes were obtained using EyeQuestion (Logic8), using unstructured line scales ranging from 0 to 100. Eight bouillons were also tested across different sessions to check for “between-session” variation to ensure the reproducibility and reliability of sensory data across multiple sessions. In total, seventy-nine bouillons were tested by multiple panelists.

In sensory studies, panelists commonly differ in their use of the assessment scale [13]. For a given attribute, panelists may exhibit level effects (consistent over- or underscoring) or scaling effects (different ranges of score usage). Also, panelists may use different ranges on the rating scale (scaling effect), e.g., one panelist scores all samples between 40 and 50 whereas another panelist assigns scores between 20 and 70. As not all bouillons were assessed by all panelists in this study (unbalanced design), both the level and the scaling differences could adversely affect the sensory data quality even after averaging the scores across panelists. Therefore, standardization was applied to the sensory data to remove level and scaling effects in an unbiased manner [14]. The effect of standardization was monitored by fitting linear mixed models using the lme4 package [15] to the data and assessing the F-value of the product effect, calculated using the lmer Test package [16] in R 4.0. After standardization, sensory scores were averaged across assessors for downstream analysis.

### 2.3. Untargeted Volatile Analysis

For the determination of the volatile levels, stir bar sorptive extraction (SBSE) was used to trap the volatiles from the liquid matrix to a polymer-based coating stir bar (polydimethylsiloxane, PDMS, 10 mm length, 0.5 mm film thickness). Analytes trapped by the stir bar were thermally desorbed (TDU, Gerstel, Mülheim, Germany) and analyzed using GC-MS (Agilent Technologies, Ratingen, Germany). Extraction and analysis followed the same procedure as described before [17], with slight modifications. In brief, bouillons were defrosted and sonicated for 10 min, and a 9 mL aliquot was pipetted into a 10 mL screw-cap vial. Immediately, a magnetic stir bar was immersed into the liquid sample. The vial was then incubated at 60 °C for 10 min and stirred at room temperature for another 80 min. After that, the stir bar was removed, rinsed with water, dried and placed into a glass desorption tube. Thermal desorption of the analytes onto the column was performed using a similar method as described before [15]; also, the same GC-MS conditions were used. The two methodological deviations from the previous study were that samples were desorbed in splitless mode during the first two minutes of the CIS parameters, and the temperature used to desorb the analytes into the GC, as well as from the GC column to the MS, was set to 270 °C.

The analysis sequence was as follows: Seventy-nine bouillon samples were analyzed in a randomized way. Duplicates of twelve of the samples were also prepared to check for repeatability. A quality control (QC) sample, which was a mix of all bouillons, was also repeatedly analyzed along the sequence after every ten samples to test the performance of the method. An empty glass tube, a clean stir bar and a blank stir bar (water) were also measured at the beginning of the sequence to check for background peaks. An n-alkane series (C_8_–C_22_) was analyzed to calculate retention indices (RIs). All samples were analyzed in a single series, but the bouillons were prepared and extracted in a set of eight batches of 15 samples each before they were placed in the GC rack. This was done to avoid long waiting times on the autosampler. The whole analysis sequence took around 5 days.

The raw GC-MS data were processed using an untargeted metabolomics approach as detailed before [15]. In brief, raw data were baseline-corrected mass peaks of samples were aligned using MetAlign software (version 0.12.5) and mass spectra were reconstructed using MSClust (version 5.4.6). Metabolites were identified by matching the mass spectra and RIs to authentic reference standards or those in the NIST17 Mass Spectral library (v.2.3), following the MSI criteria for metabolite identification as proposed by Sumner et al. [18]. Compounds that did not fit the criteria were annotated as being non-identified. Heatmaps combined with clustering analysis were used to visualize the chemical data. For the metabolite heatmap, data were transformed (sqrt), rescaled to an interval of [0, 1], and clustered using the Ward criterion (R 4.0). The sensory heatmap was prepared in analogy, but without the initial transformation step.

### 2.4. Untargeted and Targeted Non-Volatile Analysis

For the determination of non-volatile metabolite levels, an untargeted RPLC-MS method was used. Chicken bouillon samples were prepared for RPLC-MS analysis using a protein precipitation method, adding 900 µL of methanol to 200 µL of chicken bouillon. To ensure the homogeneity of the chicken bouillon, each sample was vortexed before pipetting, and the lower end of the pipette tip was cut off to prevent clogging. For precipitation, the sample was then shaken and spun down, and 900 µL of supernatant was transferred and divided into two aliquots. Each sample was evaporated to dryness overnight at 43 °C. Afterward, the samples for RP analysis were reconstituted in 25 µL of MilliQ water (Millipak Express 40 Filter 0.22 µm, Merck, Rahway, NJ, USA) and 25 µL of acetonitrile (Ultra LC-MS, Actuall Chemicals, Randmeer, Oss, The Netherlands)) containing diclofenac, prednisolone and mycophenolic acid as non- endogenous internal standards (ISTDs).

HILIC samples were reconstituted in 50 µL of 50% MeOH in water, containing succinic acid d3 (negative mode only), cimetidine, ranitidine, amiloride, MES, MOPSO, ACES and Methyl-histidine as non-endogenous internal standards.

A quality control (QC) sample consisting of a mixture of all samples pooled together was made. After preparation, samples were randomized and run in 2 batches, positive and negative mode, which included QC samples, sample replicates and blanks. QC samples were analyzed every 10 samples and were used to assess data quality and monitor the instrument response.

A Waters AccQ-Tag C18, 100 mm column 2.1 µm was used for the separation in positive and negative mode for RPLC (gradient in Appendix A) with mobile phase A consisting of MQ water + 0.1% formic acid and mobile phase B consisting of acetonitrile + 0.1% formic acid, running at a column temperature of 60 °C and an autosampler temperature of 12 °C.

For the HILIC analysis, the Waters BEH Amide 100 mm × 2.1 mm 1.7 µm column was used, both for positive and negative mode (gradient in Appendix A) with mobile phase A consisting of 90% acetonitrile + 0.1% formic acid and ammonium formate and mobile phase B consisting of 10% acetonitrile + 0.1% formic acid and ammonium formate, running at a column temperature of 45 °C and an autosampler temperature of 12 °C.

All measurements of the study samples for LC-MS were performed on a Sciex X500R QTOF mass spectrometer with ESI ionization with a mass range from 50–1200 *m*/*z*. The QTOF was calibrated every 2 h during measurements.

Targeted HILIC analysis was performed, and standard mixes were injected at the start and end of the run for target identification. After peak integration with Sciex OS (version 1.7), each target in the QC pools was corrected with each ISTD to find which was optimal. The optimal ISTD was defined as the one giving the lowest RSD of the target in the QC pools after correction.

For the untargeted non-volatile analysis, raw data files (Sciex wiff2) were converted without centroiding to mzML using MSConvert [19] (v. 3.0, ProteoWizard), cutting off data collected before 0.8 min and after 8.5 min. The MSnbase [20] package in R [21] was used to smooth and centroid the data, and the XCMS package [22] in R was applied for peak detection, (retention time) alignment and correspondence (grouping features over samples). Appendix A summarizes the parameters used. Missing values (intensities) were partially recovered using the fillChromPeaks method. The remaining missing values were replaced by half of the minimum intensity for that feature over all samples. The CAMERA package [23] in R was used to annotate isotopes, define PCgroups and annotate potential adducts/fragments. The resulting csv file was further filtered in Excel (Microsoft) to exclude, e.g., isotopes, features that are also high in blanks and features with high variability in QC pools. More detailed information on the filtering is included in Appendix A.

### 2.5. Variable Selection and Predictive Models

To explore the relationships between sensory attributes and the chemical composition of the bouillon samples, the MUVR R package [9] was used with a Random Forest regressor (R 4.0).

MUVR uses a double cross-validation approach [24] to obtain an unbiased estimate of prediction performance and an unbiased set of selected variables. By repeatedly applying a recursive variable elimination procedure, MUVR assigns importance scores to each variable and defines subsets of relevant variables. In addition to a minimal optimal (*min*) set of variables, MUVR also provides an all-relevant (*max*) variable set to enable a more complete view of which compounds are relevant for the respective sensory attribute and a *mid* variable set that represents a compromise between performance (*min*) and comprehensiveness (*max*).

For identification of relevant variables, MUVR Random Forest regression models were fitted on the GC, RPLC and HILIC datasets separately as well as on a combination of all three datasets (10 outer cross-validation folds, 200 repetitions) for each sensory attribute using the metabolomic data as predictors (biological and technical replicates aggregated into means). The R^2^ values reported by MUVR, corresponding to the out-of-bag predictive performance of the Random Forest models in the inner cross-validation loop, were used to measure the predictive performance of each model.

## 3. Results

### 3.1. Discriminative Performance of the Sensory Panel

Sensory descriptive analyses were carried out to obtain a broad sensory profile of the set of bouillon samples, including attributes related to odor, taste, aftertaste, afterfeel and mouthfeel. In total, 34 attributes were rated by trained panelists on a scale between 0 and 100. Eight sensory panel sessions were held; for each, 11–14 panelists were present. In total 15 panelists performed the assessments; some panelists were only present for three sessions, and the rest of the panelists were present between six and eight times. However, all products were tested by 11–14 panelists. With this design, it was ensured that each panelist tested a large number of samples. Raw sensory data were standardized as explained in Section 2.2, resulting in a more significant product effect for most sensory attributes in a linear mixed model (Appendix A), suggesting that the removal of panelist-specific effects enables a better view of sensory differences between products. Figure 1 shows the F-value and the significance of the product effect for each sensory attribute after standardization, with a high F-value indicating that the sensory panel could clearly distinguish the products with regard to the respective sensory attribute. This is a prerequisite for using the sensory attributes as the outcome variables for downstream modeling analyses. Overall, 27 out of 34 sensory attributes are significant at the *p* < 0.01 level, broadly covering the sensory space and showing that sufficient variation in odors, flavors, mouthfeel and aftertaste is present in the experiment and was picked up by the sensory panel. In contrast, none of the afterfeel attributes are significant at the *p* < 0.01 level, indicating either that the products in the experiment did not vary significantly in afterfeel or that existing variation could not be reliably perceived by the sensory panel.

With the broad range of sensory attributes and well-designed panel assessment sessions, a wide comprehensive sensory coverage has been achieved. One key development is the broad variety of sensory attributes that have been identified and explored. This expanded list of attributes will allow for a more nuanced analysis. Moreover, a wide range of bouillon compositions has been examined, further contributing to the diversity of this study.

The score responses of the panelists were standardized per assessor as described in Section 2.2, with the aim of mitigating possible distortions of product effects caused by the unbalanced design of the sensory study, where not all products were assessed by the same set of assessors [13]. After standardization, the sensory readings were averaged across assessors, yielding the estimated product effect. The results of the standardization procedure were assessed by ANOVA, where the F-value of the product effect measures how well the food samples could be distinguished by the assessors based on the respective attribute (Figure 1). We observed that standardization consistently increased the product effect F-value.

However, even after standardization, the F-values of some attributes remained below the critical F-value (dotted line), showing that the assessors could not detect significant differences between the products with regard to these attributes.

Some of the strongest categories by F-values were umami flavor, chicken flavor and odor. Fatty odor, yeast odor/flavor, musty od, sour flavor/aftertaste, astringent mf, mouthcoating afterfeel and astringent afterfeel can be seen as statistically irrelevant because they could not be properly distinguished by the panelists in the tested set of samples (Figure 1).

Our study revealed a great diversity in sensory attributes and showed proportionality in the dosages of the base food matrix and chicken flavor and odor. In Figure 2, all samples were sorted by their base chicken aroma dosage (low, medium, high) and plotted with the intensity measured for chicken odor and flavor, respectively, revealing a correlation between the scorings and the actual chicken aroma dosage. This proportionality was also observed across all different YP types studied (blends, process flavors and yeast extract). This indicates a consistent or proportional relationship between the food base matrix and chicken flavor and odor. It should be noted that chicken odor and flavor for yeast extract blends seem to be increasing less with higher base dosages, which might reflect a saturation effect.

### 3.2. Principal Component Analysis Quality Assurance

Principal component analysis (PCA) was used to ensure the quality of data and to exclude shifts or inconsistencies (Figure 3). Several distinct groupings within the various YPs can be seen. Generally, the types of YPs (blends, process flavors and yeast extracts) are grouped together, especially in the mass spectrometry PCAs, and show differences based in their base chicken bouillon concentrations. Therefore, the base bouillon content delivers only a small amount of variation in our analytical methods, and altogether, the variation in our samples is more defined by the added YPs than by the base chicken content.

### 3.3. Pearson Correlation of Taste Attributes

Figure 4 shows the pairwise Pearson correlations between all sensory attributes. As can be observed, scores for certain sensory attributes are highly correlated. For example, odors and flavors from the same attribute are positively correlated (e.g., salt flavor and salt afterfeel; roast flavor and roast odor). Next to that, attributes such as balance, umami, salt, sweet, intensity, length, complexity and fullness are highly correlated, meaning that some bouillon samples scored high in these seven sensory attributes simultaneously. In addition, a correlation between chicken flavor and odor and fatty mouthfeel can be observed (Figure 4), which is interesting when looking for the highly desired chicken aroma and taste characteristics for bouillons. The same can be seen for roast and bitter attributes and the yeast attributes to a lesser extent. It makes sense that fatty odor and mouthfeel and mouthcoating afterfeel correlate highly.

### 3.4. Random Forest Regression Performance—Modeling the Link Between Sensory and Chemical Profiles

To identify metabolomic features that are related to the variation in the sensory profiles of the bouillons, MUVR was applied (Figure 5) to the RPLC-, GC- and HILIC-MS datasets as well as to a combination (concatenation) of all three datasets.

Figure 5 shows an overview of the obtained R^2^ values as measures of how well each of the 34 sensory attributes could be modeled using the dataset. Sensory attributes that show little variation between samples (e.g., with low F-values in Figure 1) tend to have worse model performance, which is expected [10]. Comparing platforms, it appears GC-MS performs better on odors, and LC-MS and HILIC-MS perform better on flavors. Overall, umami, salt, intensity flavor and length aftertaste were predicted best. Some attributes could be well modeled including roast, chicken, sulfury and off-odor odor and roast, chicken, herbs, sulfury and balance flavor by the volatile profiles while others could not be—such as yeast and musty odor, astringent mouthfeel or astringent afterfeel. Specifically, HILIC-MS measurements gave strong prediction models for umami, salt, sweet intensity and length flavor and fullness mouthfeel, whereas GC-MS predicted roast odor and flavor; sulfury odor and flavor; and salt, umami, chicken, herbs and balance flavor, and RPLC-MS predicted sweet, salt and overall, umami flavor as well as intensity flavor and aftertaste, herbs flavor, complex flavor and length afterfeel. The combined model predicted intensity flavor and aftertaste, salt, umami and complex flavor, fullness mouthfeel and length aftertaste very well. Almost all categories have similar or better performance in the combined model with the exceptions of roast odor, herbs and sulfury flavor, and fullness mouthfeel, which performed only slightly worse in the combined model.

For models that incorporate features irrelevant to the predicted sensory value, the predictive quality drops. The chance of incorporating irrelevant features increases when the number of irrelevant features to choose from increases. This can be seen for the sensory attributes yeast and musty odor, where the models based on HILIC-MS have a very low R^2^ value.

### 3.5. Importance of Volatile and Non-Volatile Variables per Sensory Attribute

Besides looking at pure performance, we also looked at selected variables. We wanted to compare relevant features from each MS platform for predicting each attribute. Based on the R^2^ values of our models (Figure 5), we expected that odors would have a majority of GC-MS features selected in the combined model, while flavors and aftertastes would have more from LC-MS and HILIC. This was investigated by the predictive performance of machine learning methods, where better predictive performance corresponds to a more suitable analysis platform for the investigation of certain sensory attributes.

A view of which types of variables were selected by which model per MS platform for each attribute is shown in Figure 6. The models were made by MUVR on all datasets combined, and then the selected variables (Figure 6; Appendix A) were colored by their platform of origin and the number of features which were ranked by their importance to the MUVR-suggested min, mid or max models. These plots show some interesting differences between flavor and odor attributes. Oddly, HILIC also ranks high in some odor attributes, which comes from the correlations of flavor and odor attributes. RPLC-MS makes up 88.7% of the combined dataset (2600 variables), GC-MS 10.0% and HILIC 1.3%. HILIC features very often appear in the top five variables of min models. This is likely related to the choice of targeted compounds, which were specifically chosen to be complementary and add metabolomic coverage that GC-MS and RPLC-MS might lack. The targeted approach of HILIC is also reflected in a lower coefficient of variation (CV) in the QCs compared to RPLC-MS (Appendix A). The HILIC method was specifically tailored to its targets (sugars, polar compounds), resulting in better precision. As expected, more volatile variables appear in odor attributes and non-volatiles in flavor attributes.

Interestingly, bitter flavor and bitter aftertaste are primarily covered by measurements of volatiles. Most aftertaste attributes are covered by HILIC and LC data, but sour and bitter aftertaste are dominated by compounds measured by GC. Further looking into this example specifically shows that sour flavor is more determined by non-volatile compounds, but sour aftertaste is more determined by volatile compounds. It is possible that bitter aftertaste might occur from reactions of those compounds during tasting, similar to what was observed for wine aftertaste [25].

## 4. Discussion

The advanced development of metabolomics technologies and chemometric tools that are used to relate metabolites with flavor characteristics have expanded our knowledge of the processing of food ingredients [26,27,28]. Here, the sensory and chemical data obtained from a large set of bouillons with added YPs have been shown to cover sufficient variability in order to attempt to predict certain sensory characteristics from the detected patterns.

The prediction models applied here are based on multivariate regression models that have been proven valuable in revealing reliable relations between molecules and sensory attributes in several food products [29]. It was decided to generate Random Forest models.

Many flavor researchers have already noticed a co-dependence between sensory attributes on taste enhancers, such as roast and bitter [30,31]. Interestingly, studies have shown that umami enhances saltiness sensations [32], as well as sweetness as they share a common receptor subunit [33,34]. Research on kokumi sensation showed that salt, sweet and umami attributes enhance “richness” [35], which might explain the correlation to intensity flavor and aftertaste, length aftertaste, complex flavor and fullness mouthfeel in our study.

Among the selected variables being important for predicting roast odor (Table 2) are two alkylpyrazines (2-ethyl-5-methyl pyrazine and 2,3,5-trimethyl-6-isopentyl pyrazine), two pyrazine-type volatiles and four unknown compounds. 2-Ethyl-5-methyl pyrazine is known to be a sweet-, bean- and coffee-tasting compound and has been found in a number of diverse food items that are often consumed after roasting. These include cereals and cereal products, coffee and tea (www.foodb.ca, accessed on 1 July 2024). Isomers of the above-mentioned pyrazines were also found in the volatile composition of fenugreek [36]. These and other pyrazines are compounds formed by Maillard reactions and Strecker degradation [2]. Some can also be formed by heating hydroxyamino compounds such as threonine, serine, ethanolamine and glucosamine [37]. Alkylpyrazines have also been previously detected in roasted sunflower seeds [38], where the abundance of alkylpyrazines increased with an increase in roasting temperature and time. These pyrazines contributed most to the roast and nutty flavor of roasted sunflower seeds, but also contributed in many other oil seed products: sesame oil, peanut oil, perilla seed oil [39]. 2,3,5-Trimethyl-6-isopentyl pyrazine is not a common alkylpyrazine but has been reported as being formed in a model system from the reaction between L-ascorbic acid and L-leucine at a basic pH [40].

Some volatiles, such as 2,4-decadienal, known to be related to chicken aroma [6,41], were present in all chicken bouillon samples, meaning that these compounds are likely to have come from a common ingredient (the chicken aroma base) and as expected were highly present in the samples containing high levels of chicken aroma and not highly present in the sample containing low levels of chicken aroma. Cystine showed the highest correlation in chicken flavor and odor and can react with ribose to form 2-methyl-3-furanthiol which was previously described as “the most important compound in chicken flavo[u]r” [6].

Fructose-leucine was highly related to roast flavor in this study. In 2019, Guo et al. [38] reported this compound is affected by the roasting conditions of sunflower seeds. Similarly, furan-containing compounds, such as fructose-leucine, are likely thermally degraded from fructose and glucose and have been shown to be related to meat flavor [42].

Cyclo-leucine-proline, a dioxopiperazine that might be derived from a pyrazine, was linked to roast flavor and is known, next to other cyclic dipeptides, to modulate taste. Leucyl-proline was found in other studies to be the result of Maillard reactions in black garlic [43].

Glutamate, GMP and IMP are well-known umami compounds [44,45,46] that were also confirmed in our measurements to be highly ranked in the prediction of sweet, umami and salt flavor. The same is true for aspartate, which has been proven to “enhance umami, saltiness and overall flavor” [45]. Aspartylphenylalanine has been found to be related to salt flavor here and is known to be a metabolic by-product of its methyl ester, aspartame.

As also identified in our previous study [10,47], gamma-glutamyl-phenylalanine (FDB000716), associated with “kokumi taste” in the literature [10,47], was confirmed to contribute to salt flavor in chicken bouillon. Methionyl-leucine correlated to sweet flavor.

Two pyrazines were found to correlate highly with roast odor, as well as other unidentified compounds that appear to have a pyrazine-related nature. This makes the presence of these volatiles important for enhancing roast odor in chicken bouillon.

We have demonstrated the applicability and feasibility of using both targeted and untargeted data for our combined model. HILIC as a targeted method here added value in specific predicted categories, but it also reflects its limitation by not adding additional information to certain attributes such as yeast odor and astringent afterfeel. However, any other attribute prediction by the combined model was either improved or unchanged in its performance using the combined information of all platforms.

The distinction between prediction/correlation and causation is a crucial concept in data analysis and research. The process of measuring correlations, here used for predictive modeling, focuses on identifying patterns or relationships between variables without necessarily implying a cause-and-effect relationship.

The issue arises when correlations are mistakenly interpreted as causal relationships. In our approach, we gathered additional metabolomic data, which helped to confirm existing correlations and potential causal relationships. Indeed, the literature supports the connection between volatile and non-volatile metabolites, as well as between flavor and odor variables, for example.

The combined sensory–chemical approach and the proposed tools described here can be used to define sensory and chemical properties of similar processed food matrices with added YPs to monitor the flavor formation in food design using more efficient procedures. Careful attention is needed for the design of the sample products, so that sensory and chemical space is properly covered. We could also demonstrate, for each attribute, which analytical platform works best or adds more relevant data. These data can potentially be used for similar food matrices with new or improved added YPs. Predictive models might then be further applied to a larger set of food products to estimate the sensorial properties by using instrumental analysis only. This approach can be used to save some time in food design by removing the need for costly taste panels in the early stages of product design.

This study described a straightforward data-driven approach for studying foods with added YPs to identify flavor-impacting correlations between molecular composition and sensory perception. We presented a matrix-based approach for the prediction of food taste, which can be used to analyze foods for targeted flavor design or quality control.

We used an example of chicken flavor profiles, but this approach could be used in a similar manner with other food matrices.

## Figures and Tables

**Figure 1 metabolites-15-00317-f001:**
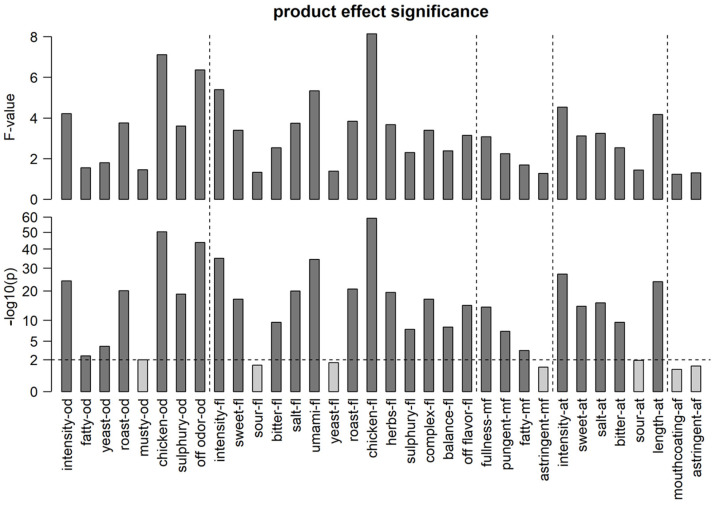
Product effect F-values (**top**) and *p*-values (**bottom**) per sensory attribute after the application of standardization, as estimated by linear mixed models. Higher F-values and lower *p*-values indicate a more significant product effect, meaning that differences between products were clearly detected by the sensory panel. The horizontal dashed line represents a *p*-value of 0.01. Abbreviations: -af: afterfeel, -at: aftertaste, -mf: mouthfeel, -fl: flavor, -od: odor.

**Figure 2 metabolites-15-00317-f002:**
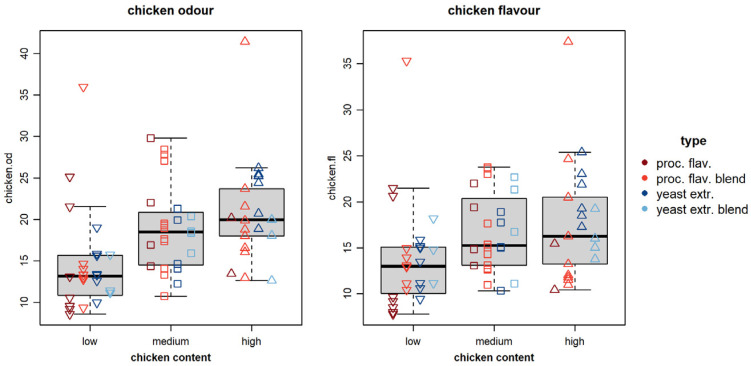
The bar diagrams demonstrate the relationship between chicken flavor and odor scorings related to the variant YPs (blue = yeast extract, teal = yeast extract blend, brown = process flavor, red = process flavor blend) and the chicken base concentration (
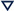
 = low, 
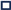
 = mid, 
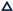
 = high).

**Figure 3 metabolites-15-00317-f003:**
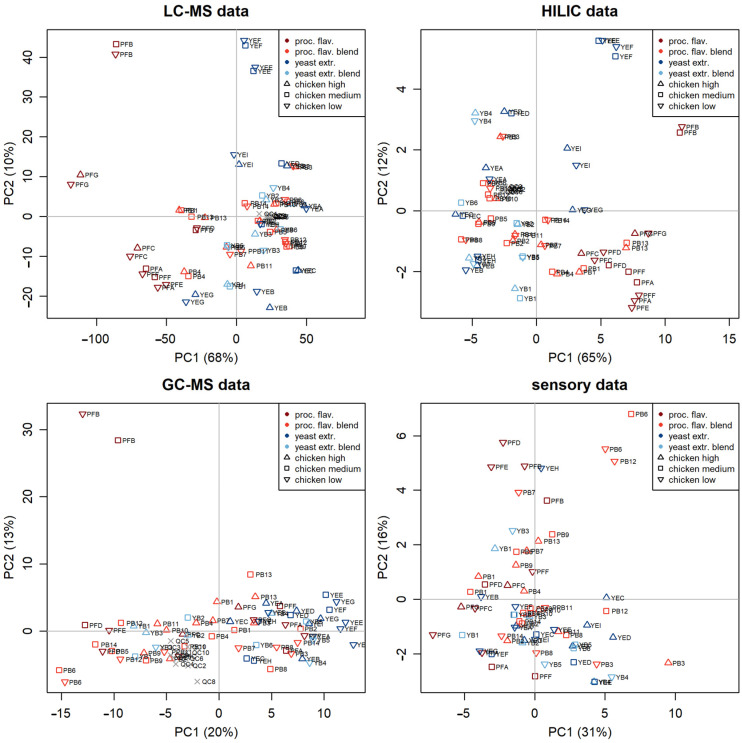
Principal component analysis (PCA) of the log-transformed, minimum-value-imputed untargeted RPLC-MS, targeted HILIC-MS, GC-MS and sensory data. Different YPs are indicated by different colors (red = process flavor blend, green = yeast extract, blue = process flavor, turquoise = blend yeast), and different chicken base concentrations are indicated by the symbols (high, mid or low). The corresponding loading plots are shown in Appendix A.

**Figure 4 metabolites-15-00317-f004:**
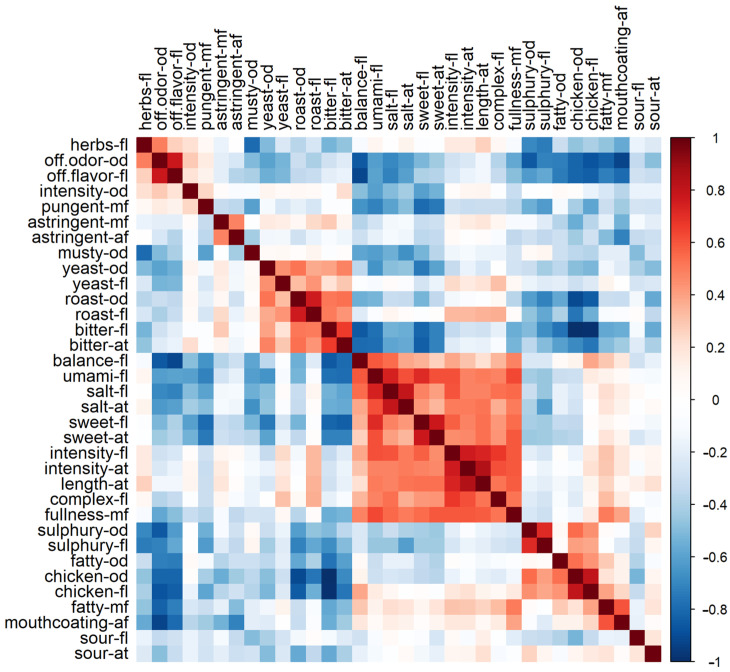
Pearson correlation matrix of the sensory chicken bouillon data showing the correlation values of the different sensory attributes (high correlation: dark red, no correlation: white, negative correlation: blue).

**Figure 5 metabolites-15-00317-f005:**
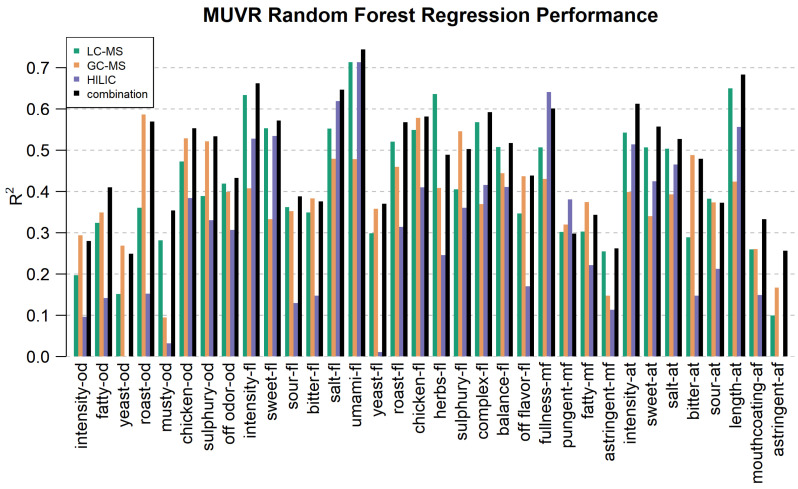
R^2^ values of the predictive model (Random Forest) for each sensory attribute and each platform (blue = HILIC-MS, yellow = GC-MS, green = RPLC-MS, black = all combined).

**Figure 6 metabolites-15-00317-f006:**
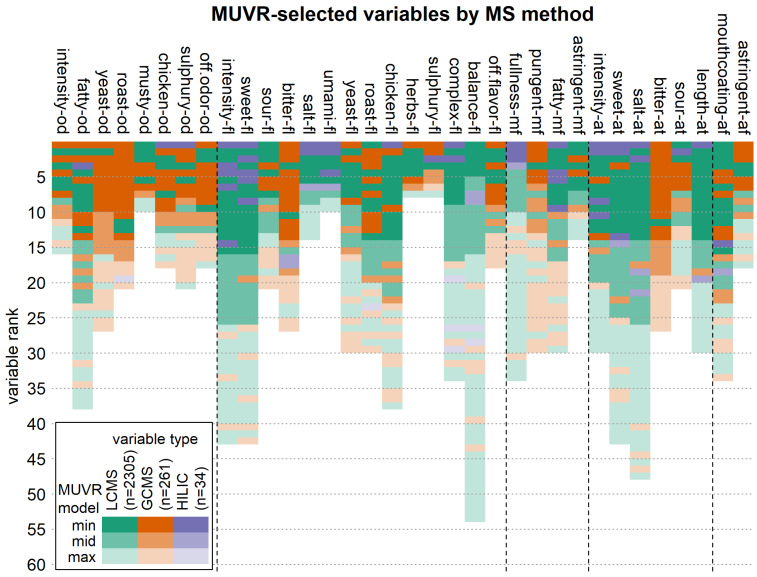
Variable importance and relevant variable subsets in the combination models. For each sensory attribute, the importance and the origin (LC-MS and GC-MS or HILIC) of the selected variables in the combined MUVR regression model are shown. Variable importance is represented as the variable rank, with lower variable ranks indicating higher variable importance. While the color indicates which dataset a variable stems from—LC-MS (green), GC-MS (orange) or HILIC (violet)—the color intensity indicates whether a variable is part of the min (minimal optimal, i.e., most relevant) variable set or belongs to the less relevant mid or max (all-relevant) sets.

**Table 1 metabolites-15-00317-t001:** Overview of the sensory attributes assessed by QDA.

Sensory Attributes					
**Odor ** **(od)**	1	intensity-od	**Mouthfeel ** **(mf)**	23	fullness-mf
	2	fatty-od		24	pungent-mf
	3	yeast-od		25	fatty-mf
	4	roast-od		26	astringent-mf
	5	musty-od	**Aftertaste ** **(at)**	27	intensity-at
	6	chicken-od		28	sweet-at
	7	sulfury-od		29	salt-at
	8	off-odor-od		30	bitter-at
**Flavor ** **(fl)**	9	intensity-fl		31	sour-at
	10	sweet-fl		32	length-at
	11	sour-fl	**Afterfeel ** **(af)**	33	mouthcoating-af
	12	bitter-fl		34	astringent-af
	13	salt-fl			
	14	umami-fl			
	15	yeast-fl			
	16	roast-fl			
	17	chicken-fl			
	18	herbs-fl			
	19	sulfury-fl			
	20	complex-fl			
	21	balance-fl			
	22	off-flavor-fl			

**Table 2 metabolites-15-00317-t002:** This is an overview of putative identifications of the highest-ranked features measured by LC-MS, HILIC-MS and GC-MS of the selected sensory attribute models. Abbreviations: -af: afterfeel, -at: aftertaste, -mf: mouthfeel, -fl: flavor, -od: odor. For more details, an extended table is included as Appendix A.

**Chicken Flavor**			**Chicken Odor**		
**Rank**	**Measurement**	**Likely Hit/Formula**	**Rank**	**Measurement**	**Likely Hit/Formula**
1	HILIC	Cystine	1	HILIC	Cystine
2	GC	1-octen-3-ol	2	GC	2,4-nonadienal
3	RPLC	C18H7N10NaO	3	RPLC	N-Fructosyl glutamylphenylalanine or isomer
4	RPLC	C19H38N3NaO8S3	4	GC	2,4-decadienal
5	RPLC	N-Fructosyl glutamylphenylalanine or isomer	5	GC	1-octen-3-ol
			6	RPLC	Traumatic acid or isomer
			7	RPLC	C18H7N10NaO
			8	GC	2,4-decadienal
**Sweet Flavor**			**Salt**		
**Rank**	**Measurement**	**Likely Hit/Formula**	**Flavor**		
1	HILIC	GMP	**Rank**	**Measurement**	**Likely Hit/Formula**
2	RPLC	C17H23NNa2O5	1	HILIC	GMP
3	HILIC	IMP	2	HILIC	IMP
4	HILIC	UMP	3	RPLC	N-gamma-L-Glutamyl-L-phenylalanine/Aspartame
5	HILIC	CMP	4	RPLC	C12H22N4O5
6	RPLC	C23H41N2O3PS	5	RPLC	Aspartylphenylalanine/Phenylalanyl-aspartic acid
7	RPLC	C16H19N0O5	6	RPLC	C13H25N5NaO2P
8	HILIC	L.Glutamic.acid	7	HILIC	UMP
9	RPLC	C15H25N3O8	8	RPLC	C16H19N0O5
10	RPLC	C15H25N3O8	**Umami Flavor**		
11	RPLC	C16H19NNa2O3	**Rank**	**Measurement**	**Likely Hit/Formula**
12	RPLC	C17H30N6O6	1	HILIC	GMP
13	GC	homosalate	2	HILIC	IMP
14	RPLC	C19H27N3O4/C14H32N3NaO4S	3	RPLC	C18H36N4O4
15	RPLC	Methionyl-Leucine or Isomer	4	HILIC	CMP
			5	RPLC	Leu-Ala-Ser, Ala-Val-Thr or Thr-Gly-Leu or isomer
			6	HILIC	UMP
			7	RPLC	C19H27N3O6
**Roast Flavor**			**Roast Odor**		
**Rank**	**Measurement**	**Likely Hit/Formula**	**Rank**	**Measurement**	**Likely Hit/Formula**
1	RPLC	Fructose-isoleucine, Fructose-leucine or isomer	1	GC	C14H20O2
2	GC	non-identified	2	GC	non-identified
3	GC	4-methoxy-6-(2-propenyl)-1,3-benzodioxole	3	GC	C12H11N
4	RPLC	C22H15N4NaO5	4	GC	C11H18N2
5	GC	4-terpineol	5	GC	non-identified
6	RPLC	C9H15NO5	6	GC	C10H16N2
7	GC	C11H18N2	7	GC	C7H10N2
8	GC	non-identified	8	GC	non-identified
9	RPLC	C20H7N3O5P2	9	GC	2-ethyl-5-methyl pyrazine
10	GC	terpineol	10	RPLC	Fructose-isoleucine, Fructose-leucine or isomer
11	RPLC	C33H24N2	11	GC	2,3,5-trimethyl-6-isopentyl pyrazine
12	RPLC	C11H21N3O2	12	RPLC	non-identified
13	RPLC	non-identified			
14	GC	non-identified			
15	RPLC	L,L-Cyclo(leucylprolyl)			
16	RPLC	non-identified			
17	GC	2,3,5-trimethyl-6-isopentyl pyrazine			

## Data Availability

The data presented in this study are openly available at https://doi.org/10.4121/f1fca011-141c-4994-989f-b98f44ceae5f and https://doi.org/10.4121/3b3107f5-b58c-4b2b-a4b1-973530d0f18e (accessed on 17 March 2025).

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
