# Peer review of "A Data-Driven Approach to Link GC-MS and LC-MS with Sensory Attributes of Chicken Bouillon with Added Yeast-Derived Flavor Products in a Combined Prediction Model"

_metabolites, 2025, doi:10.3390/metabo15050317_

Round 1

Reviewer 1 Report

Comments and Suggestions for Authors

Dear authors,

Thank you for your enormous effort in finishing this study, entitled,” A Data-Driven Approach to Link GC-MS and LC-MS with Sensory Attributes of Chicken Bouillon with Added Yeast-Derived Flavor Products in a Combined Prediction Model”. Following a thorough peer reviewing process, the reviewer inferred that this study may provide sufficient information and quality to be published in Metabolites. However, plenty of revisions are required to improve the significance of the proposed content. Although the proposed idea was potential, Background study needs significant improvements. Especially the research gap and what will author proposes from the study is not well written/shown. Therefore, the reviewer suggests that the author could provide some more paragraphs that explain that essential matter, previous study related to this topic. Most of the materials and methods were not in a clear explanation, which require improvement in writing. In addition, to ensure possible reproducibility, each parameter should be written with the cited protocol or references.

Sincerely,

Comments on the Quality of English Language

The English could be improved to more clearly express the research. Please do professional English editing.

Author Response

Thank you for your recommendation and your effort to help improve our manuscript!

I hope we could address your revisions in a sufficient way listed below and implemented in the manuscript.

  • The abstract, while descriptive, could benefit from a more concise and impactful summary of the study's key findings. Rephrase to highlights the most significant correlations discovered and the predictive power of the models developed. For example, mention the key volatiles associated with roast and chicken odors, and quantify the performance of the Random Forest models in predicting these attributes.
  • The abstract doesn't explicitly state the main conclusion or the potential applications of the research. Add a concluding sentence that summarizes the main takeaway and suggests future directions or applications, such as using the models for targeted flavor design or quality control.

Remark 1 and 2 are implemented and marked in the manuscript. The abstract has been edited to emphasize impact and potential applications.

  • The statement "...This is the first example of such an extensive study..." might be an overclaim. Temper the statement to: "This study represents an extensive effort combining a professional sensory panel with multi-platform metabolomics to investigate the complex flavor profiles of bouillons with compositional variations."

Implemented and marked in the manuscript (ll.136).

  • The sentence describing yeast-derived products (YP) could be clearer. Rephrase to: "Yeast-derived products (YP) offer several advantages, including the potential to replace artificial flavorings while enhancing flavor complexity and balance."

Implemented and marked in the manuscript (ll.88).

  • The statement about limited sensory variability impacting prediction models is vague. Elaborate: "This pilot study revealed that prediction models struggle when samples exhibit limited sensory variation, highlighting the need for broader compositional diversity."

Implemented and marked in the manuscript (ll.116).

  • Specifying Unilever as the source of chicken bouillon could be perceived as biased. Generalize: "Commercial chicken bouillon powders were used as a base matrix."

Implemented and marked in the manuscript (ll.142).

  • More details needed on "low" chicken aroma dosage. Quantify: "..."low," which contained X% less chicken aroma than the "middle" dosage."

The dosages of the base chicken bouillon are compared to commercial formulations (high= 80%, middle= 40%, low= 0%). This has been implemented and marked in the manuscript (ll.145).

  • The phrase "standard protocols" is vague. Specify which sensory evaluation standards were followed (e.g., ASTM E253-14).

Implemented and marked in the manuscript (ll.175, ref. 11). ISO 6658:2017 Available online: https://www.iso.org/standard/65519.html (accessed on 7 April 2025).

  • Define what constitutes an "experienced, trained" panelist. Include details on training duration, frequency, and qualification criteria.

Implemented and marked in the manuscript (ll.180).

  • Explain the rationale behind checking for "between-session" variation. Clarify why this check was important (e.g., "to ensure the reproducibility and reliability of sensory data across multiple sessions").

Implemented and marked in the manuscript (ll.190).

  • Simplify and clarify the explanation of panelist effects. Rephrase: "Panelists may exhibit level effects (consistent over- or underscoring) or scaling effects (different ranges of score usage)."

Implemented and marked in the manuscript (ll.195).

  • The description of SBSE is concise but could benefit from mentioning the coating thickness. Include: "...stir bar (polydimethylsiloxane, PDMS, X μm coating thickness)."

Implemented and marked in the manuscript (ll.210).

  • Generally, the Results section lacks a strong narrative flow. It presents findings in a somewhat disjointed manner without clearly connecting them to the research questions or highlighting the most important discoveries. Organize the results to tell a more coherent story. Prioritize the most significant findings and present them in a logical sequence that leads to the key conclusions. Emphasize the relationships between different datasets (sensory, GC-MS, LC-MS) and how they contribute to the overall understanding of the flavor profile. Use subheadings to break up the text and guide the reader through the different analyses.

The entire manuscript was reread with this comment in mind. Headings were reworded to better indicate key points and the organization was adjusted to tell a more coherent story.

  • Figure 2 & others Need to include error bars. In order to indicate the distribution and validity of your data, please include the error bars.

Figure 2 includes error bars and other figures would not really benefit from error bars. The other figures don’t show distribution values, but single value modelling parameters, therefore we included the coefficients of variation of the model in Supplementary Figure 5 to reflect its precision instead.

  • Table 2. Too much information. Please show only the relevant result that helps in discussion.

The table was simplified and the extended table was moved to the supplementary data, this improves the readability and the complete information is still available.

  • Figure 5 The figure did not mention the software. Please add what software did you use to generate the graph.

Method section mentions that figures are generated by a Random Forest regressor (R 4.0, ll.314).

  • The discussion does not highlight if your machine learning can predict any flavor attributes in real food products. Please do pilot trial to measure the robustness of your results.

As both reviewers acknowledge this is already an complex, broad and extensive study. We would like to implement our models on real food products, however this is not possible at this time and we hope this publication would help justify using this approach in a future project.

Reviewer 1 indicated the English could be improved to more clearly express the research

A native English speaker reviewed the manuscript to help identify where the writing could be improved

Reviewer 2 Report

Comments and Suggestions for Authors

The manuscript entitled “A Data-Driven Approach to Link GC-MS and LC-MS with Sensory Attributes of Chicken Bouillon with Added Yeast-Derived Flavor Products in a Combined Prediction Model” is devoted to the flavor and odor analysis of yeast products with multiple methods and random forest algorithms.

In the field, there is an ongoing need to develop innovative and enhanced sensory food experiences. In the food industry, flavor products derived from yeast are commonly utilized as ingredients to introduce new aromas and taste profiles that consumers enjoy. In this study, chicken bouillon samples containing various yeast-derived flavors were analyzed both chemically and sensorially using advanced statistical techniques. To identify the flavor compounds present, multiple analytical methods were employed, including stir bar solid-phase microextraction coupled with gas chromatography-mass spectrometry, non-targeted reversed-phase liquid chromatography-mass spectrometry, and targeted hydrophilic interaction liquid chromatography-mass spectrometry. Using Random Forest modeling, specific volatiles linked to roasted and chicken-like aromas were identified. The study also found several specific metabolites that were connected to the particular flavors, such as glutamate and certain peptides, characteristic to umami flavors. Overall, this research presents a data-driven methodology for examining foods enhanced with yeast-derived flavors, aiming to uncover the relationships between chemical composition and sensory experiences, while also addressing the challenges and prerequisites for developing effective predictive models.

The study is well-constructed and clearly written. There are only few suggestions that may be applied prior to its publication:

1) “Min, mid and max models” can be revised as “MUVR-suggested min, mid or max models” to ease comprehension by the readers. The terms were explained in the experimental section but it will be good to remined their meanings somewhere in the discussion and/or results, not only Fig.6 caption.

2) Table 2 looks too technical. It can be reworked somehow into a Figure to enable more clear comprehension by the readers (or can be simply moved to the supplementary file). Also, there is a question, why some features in the devoted table have property missing in the relevance column, e.g. SBSE3053 has no chicken-od, SBSE11287 and N2221 have no roast-od, P428 has no roast-fl? This should be explained in the text.

3) Table 1 can be also reworked as a diagram series (nut not strictly necessary, up to the authors)

4) Mean CV values from Fig.S5 can be listed or partially mentioned in the main text.

5) SBSE should be introduced when first mentioned in text.

6) “Sample was shaken, spun down and…” either “900 uL supernatant was” or “900 uL of supernatant were”

Author Response

Reviewer 2

Thank you for your recommendation and your effort to help improve our manuscript!

I hope we could address your revisions in a sufficient way listed below and implemented in the manuscript.

1) “Min, mid and max models” can be revised as “MUVR-suggested min, mid or max models” to ease comprehension by the readers. The terms were explained in the experimental section but it will be good to remined their meanings somewhere in the discussion and/or results, not only Fig.6 caption.

Implemented and marked in the manuscript (ll.437).

2) Table 2 looks too technical. It can be reworked somehow into a Figure to enable more clear comprehension by the readers (or can be simply moved to the supplementary file). Also, there is a question, why some features in the devoted table have property missing in the relevance column, e.g. SBSE3053 has no chicken-od, SBSE11287 and N2221 have no roast-od, P428 has no roast-fl? This should be explained in the text.

The table in the manuscript has been simplified for readability and the full list has been moved to the supplementary data. But for completion we would like to address your remarks about 4 features.

-SBSE3053 is listed for chicken-od

-SBSE11287 ranked high but not for the min model for roast-od

-N2221 is in the min model for roast-fl and high ranking, but not in the min model for roast-od

-P428 is in the min model for roast-od and high ranking, but not in the min model, for roast-fl

3) Table 1 can be also reworked as a diagram series (nut not strictly necessary, up to the authors)

We prefer the table format.

4) Mean CV values from Fig.S5 can be listed or partially mentioned in the main text.

Supplementary Figure 5 is mentioned in “3.5 Importance of Volatile and Non-Volatile Variables per Sensory Attribute” to reflect upon the models precision.

5) SBSE should be introduced when first mentioned in text.

Implemented and marked in the manuscript (ll.32;208).

6) “Sample was shaken, spun down and…” either “900 uL supernatant was” or “900 uL of supernatant were”

Implemented and marked in the manuscript (ll.255).

Reviewer 1 indicated the English could be improved to more clearly express the research

A native English speaker reviewed the manuscript to help identify where the writing could be improved
